# Low Glucose plus β-Hydroxybutyrate Induces an Enhanced Inflammatory Response in Yak Alveolar Macrophages via Activating the GPR109A/NF-κB Signaling Pathway

**DOI:** 10.3390/ijms241411331

**Published:** 2023-07-11

**Authors:** Jiancheng Qi, Qiyuan Yang, Qing Xia, Fangyuan Huang, Hongrui Guo, Hengmin Cui, Yue Xie, Zhihua Ren, Liping Gou, Dongjie Cai, Maqsood Ahmed Kumbhar, Jing Fang, Zhicai Zuo

**Affiliations:** Key Laboratory of Animal Disease and Human Health of Sichuan Province, College of Veterinary Medicine, Sichuan Agricultural University, Chengdu 611130, China; jianchengqi1992@foxmail.com (J.Q.); yqy979531@163.com (Q.Y.); xiqing9523@163.com (Q.X.); yuan1221@163.com (F.H.); guohongrui@sicau.edu.cn (H.G.); cuihengmin2008@sina.com (H.C.); zhandegaokandey123@163.com (Y.X.); zhihua_ren@126.com (Z.R.); glping0827@163.com (L.G.); dongjie_cai@sicau.edu.cn (D.C.); maqsood9040@gmail.com (M.A.K.)

**Keywords:** yak, primary alveolar macrophages (AMs), β-hydroxybutyrate (BHB), low glucose, pro-inflammatory response

## Abstract

Yaks are often subject to long-term starvation and a high prevalence of respiratory diseases and mortality in the withered season, yet the mechanisms that cause this remain unclear. Research has demonstrated that β-hydroxybutyrate (BHB) plays a significant role in regulating the immune system. Hence, we hypothesize that the low glucose and high BHB condition induced by severe starvation might have an effect on the pro-inflammatory response of the alveolar macrophages (AMs) in yaks. To validate our hypothesis, we isolated and identified primary AMs from freshly slaughtered yaks and cultured them in a medium with 5.5 mM of glucose or 2.8 mM of glucose plus 1–4 mM of BHB. Utilizing a real-time quantitative polymerase chain reaction (RT-qPCR), immunoblot assay, and enzyme-linked immunosorbent assay (ELISA), we evaluated the gene and protein expression levels of GPR109A (G-protein-coupled receptor 109A), NF-κB p65, p38, and PPARγ and the concentrations of pro-inflammatory cytokines interleukin (IL)-1β and IL-6 and tumor necrosis factor (TNF)-α in the supernatant. The results demonstrated that AMs exposed to low glucose plus BHB had significantly higher levels of IL-1β, IL-6, and TNF-α (*p* < 0.05) and higher activity of the GPR109A/NF-κB signaling pathway. A pretreatment of either pertussis toxin (PTX, inhibitor of GPR109A) or pyrrolidinedithiocarbamic (PDTC, inhibitor of NF-κB p65) was effective in preventing the elevated secretion of pro-inflammatory cytokines induced by low glucose plus BHB (*p* < 0.05). These results indicated that the low glucose plus BHB condition would induce an enhanced pro-inflammatory response through the activation of the GPR109A/NF-κB signaling pathway in primary yak AMs, which is probably the reason why yaks experience a higher rate of respiratory diseases and mortality. This study will offer new insight into the prevention and treatment of bovine respiratory diseases.

## 1. Introduction

Yaks are a fundamental part of the Qinghai–Tibet Plateau, providing essential ecological balance, economic sustenance, and the preservation of cultural heritage for this region [1]. Due to the distinct geographical and meteorological conditions, yaks are often plagued by extreme hunger and a high risk of respiratory diseases and mortality in withered seasons [1,2], the mechanisms of which are still indistinct. Although there are currently no reports on the blood glucose and β-hydroxybutyrate (BHB) levels of yaks under grazing conditions in withered seasons, a few studies have revealed that seven days of starvation would result in a rapid decrease in the blood glucose (from 3.84 mM to 3.19 mM) and an increase in the serum BHB (from 0.26 mM to 0.39 mM) level [3,4,5], indicating that grazed yaks might experience a more significant decrease in glucose and increase in BHB. Our prior research revealed that 9 days of starvation combined with 3 days of BHB injection had a major impact on the composition and structure of the nasopharyngeal microbial community of yaks, and it also raised the relative abundance of pathogens [6]. These results suggest that the combination of low glucose and high BHB levels may have an impact on the immunity of the yaks’ airways. The alveolar macrophages (AMs) are the predominant innate immune cells in the lower respiratory tract, situated on the interior of the alveolar space. They are the initial cells to come in contact with foreign pathogens and pollutants and assist in the commencement and completion of the immune response in the lungs [7]. Therefore, the dysfunction of AMs in the lungs of yaks might be responsible for their increased risk of respiratory diseases and mortality. However, this assumption needs more support.

BHB is a key component in the energy cycle, and it also has a role in moderating the intensity of inflammatory responses through various approaches [8,9,10]. It has recently been recognized that BHB plays an important role in the respiratory immune system [11]. For example, Christoph et al. discovered that BHB was beneficial to CD4^+^ T cells in the lungs, as it improved the survival rate of the cells and increased the production of interferon-γ [12]; Akiko et al. also demonstrated that a ketogenic diet activated defensive γδ T cell responses against influenza virus infection and augmented the barrier function of the lungs in mice [13]. The G-protein-coupled receptor (GPR109A) is a seven-fold transmembrane receptor that is highly expressed in adipose tissue and immune cells, and it has the ability to sense various extracellular signals, regulating cell signaling related to energy and lipid metabolism, as well as immunity [14]. As the main receptor of BHB, GPR109A participates in the regulation of the phenotype and function of macrophages [15], and GPR109A has been found to be the receptor of BHB in many reports about the effects of BHB on macrophages and immune cells [15,16]. It also has been found that BHB has the capacity to activate inflammatory signaling pathways, such as the p38/NF-κB signaling pathway [17,18], and increase the release of pro-inflammatory cytokines, including interleukin (IL)-1β, IL-6, and tumor necrosis factor (TNF)-α, in bovine hepatocyte endometrial cells [18,19]. However, some other studies revealed the anti-inflammatory properties of BHB on the M1 macrophages and monocytes [10,15]. This contradiction could be attributed to the difference in the target cells and the concentrations of BHB. In addition, low glucose was found to stimulate the activation of the NF-κB signaling pathway in monocytes, leading to an increase in the production of pro-inflammatory cytokines, such as IL-1β, IL-6, IL-8, and TNF-α [20,21]. However, it is uncertain whether BHB will have an impact on the secretory functions of yaks’ AMs in a low-glucose condition.

Hence, we hypothesize that low glucose and BHB can affect the inflammation response of yaks’ AMs through the GPR109A/p38/NF-κB signal pathway. To validate this hypothesis, we isolated primary AMs from freshly slaughtered yaks and stimulated them with low glucose and various concentrations of BHB. Subsequently, we accessed the secretory capacities of the pro-inflammatory cytokines and the activities of the GPR109A/p38/NF-κB signal pathway. This investigation will provide further insight into the underlying causes of the heightened risk of respiratory diseases and mortality of yaks during withered seasons.

## 2. Results

The isolated cells exhibited typical features of AMs, including large and round or oval shapes (Figure 1A). The results of a-naphthyl acetate esterase staining and trypan blue exclusion revealed that the majority (>97%) of the leaving adherent cells were macrophages. In order to assess the ideal concentration and duration of BHB exposure to the AMs, we administered 1, 2, 4, 8, 16, and 32 mM of BHB to the AMs for a period of either 12 or 24 h under normal glucose (5.5 mM) conditions. And the results indicated that there were no significant changes (*p* > 0.05) in the cell viability of AMs when they were exposed to 1, 2, and 4 mM of BHB for 12 and 24 h. However, when exposed to over 8 mM of BHB for either 12 or 24 h, the viability of AMs was significantly decreased (*p* < 0.05; Figure 1B). Consequently, for the subsequent experiment, we opted for 1, 2, and 4 mM BHB concentrations and 12 h of exposure duration.

### 2.1. Effect of Low Glucose plus BHB on the Secretion of Pro-Inflammatory Cytokines

To investigate the influence of BHB on the inflammatory response of AMs, we evaluated the relative expression levels of the *p65* gene and the supernatant concentrations of IL-1β, IL-6, and TNF-α. The expression level of the *p65* gene and the concentration of TNF-α in the AMs exposed to low glucose (2.8 mM) alone were found to be significantly higher than those exposed to normal glucose (Figure 2A,D; *p* < 0.05). BHB treatment had a dose-dependent effect on the expression level of the *p65* gene and the secretion of IL-1β, IL-6, and TNF-α in both low-glucose and normal-glucose conditions (Figure 2; *p* < 0.01). It is noteworthy that AMs exposed to low glucose had significantly higher *p65* expression levels and concentrations of IL-1β, IL-6, and TNF-α compared to those exposed to normal glucose in a certain BHB concentration (Figure 2; *p* < 0.05).

### 2.2. Effects of Low Glucose plus BHB on the GPR109A/p38/NF-κB Signaling Pathway

To determine whether the pro-inflammatory effects of low glucose plus BHB involves the GPR109A/p38/NF-κB signaling pathway, we accessed the gene and protein expression levels of GPR109A, p38, and PPARγ. Results from RT-qPCR revealed that, both in normal- and low-glucose conditions, BHB increased the relative mRNA expression levels of gene *GPR109A* and *p38* in a dose-dependent manner (*p* < 0.05), and low glucose augmented the BHB-induced overexpression of *GPR109A* (*p* < 0.05; Figure 3A,B). The expression level of *PPARγ* remained unchanged when exposed to BHB in normal-glucose conditions (*p* > 0.05); however, when exposed to 4 mM of BHB in low-glucose conditions, the expression level of *PPARγ* significantly decreased (*p* < 0.05). In addition, when exposed to 2 and 4 mM of BHB, the expression level of *PPARγ* in low-glucose conditions was notably diminished than that under normal-glucose conditions (*p* < 0.05, Figure 3C). When the AMs were exposed to 4 mM of BHB, GPR109A and p65 had significantly increased protein expression levels (*p* < 0.05), whereas PPARγ had significantly decreased protein expression levels (*p* < 0.05). These changes were further augmented when subjected to low glucose (*p* < 0.05; Figure 3D–F,H). Similarly, the ratio of p-p38 to p38 was also elevated in AMs exposed to 4 mM of BHB (*p* < 0.05) and further increased when exposed to low glucose (*p* < 0.05; Figure 3G).

### 2.3. Effect of GPR109A/p38/NF-κB Signaling Pathway Inhibition on Inflammatory Response of AMs Induced by Low Glucose plus BHB

To determine if the GPR109A/p38/NF-κB signaling pathway is necessary for the pro-inflammatory response induced by low glucose plus BHB, we employed PTX (an inhibitor of GPR109A) and PDTC (an inhibitor of NF-κB) to block the GPR109A/p38/NF-κB signaling pathway. As Figure 2D and Figure 4A–D show, 200 ng/mL of PTX and 10 mM of PDTC treatment effectively inhibited the enhanced expressions of GPR109A and p65 (*p* < 0.05) caused by low glucose plus BHB. However, the mRNA expression levels of them were still higher than those in the control group (*p* < 0.05). The treatment of PTX and PDTC was also found to be effective in inhibiting the overexpression of p38 and the increase in p-p38/p38 caused by low glucose plus BHB (*p* < 0.05; Figure 4E,F), as well as in increasing the expression level of PPARγ (*p* < 0.05; Figure 4G,H).

Finally, we measured the release of pro-inflammatory cytokines, such as IL-1β, IL-6, and TNF-α, in the AMs of each group. The results indicated that PTX and PTDC treatment effectively inhibited the increase in IL-1β, IL-6, and TNF-α concentrations induced by low glucose plus BHB (*p* < 0.05). However, the concentrations of IL-1β, IL-6, and TNF-α were still significantly higher than those in the control group (*p* < 0.05; Figure 5A–C).

## 3. Discussion

Yaks are often subject to severe starvation and a high risk of respiratory diseases and mortalities due to the lack of grass during withered seasons (October to May) [1,2]. It is documented that yaks experience a 15–30% decrease in body weight and a 10–20% mortality rate in cold seasons, which has a major economic impact [2]. However, the underlying mechanisms of this high mortality rate have barely been investigated. In this study, we obtained AMs from freshly slaughtered healthy yaks, assessed the inflammatory response of the AMs in a low glucose plus BHB condition, and then explored the underlying molecular mechanisms involved.

The serum levels of glucose and BHB of cows subjected to severe negative energy balance might reach 2.05 mM and over 3.0 mM, respectively [22]. We assumed that yaks suffering extreme starvation might also experience similar hypoglycemia and hyperketonemia. Hence, to simulate the low glucose and elevated BHB concentrations that yaks experience in cold seasons [2], we treated the isolated primary AMs with 2.8 mM of glucose plus 1–4 mM of BHB. The results showed that, in a low-glucose environment, BHB increased the gene and protein expression levels of NF-κB p65 and promoted the release of pro-inflammatory cytokines in a concentration-dependent manner (Figure 2). Then, subsequent experiments demonstrated that low glucose plus BHB could activate the GPR109A/NF-κB pro-inflammatory signaling pathway (Figure 3), supporting those studies that revealed that BHB regulated the function of immune cells by activating the GPR109A [15,16]. In support of our results, previous studies have demonstrated that BHB can stimulate the release of pro-inflammatory cytokines from multiple cells in cattle, including endometrial cells, hepatocytes, and neutrophils [18,19,23]. Animal research has also revealed that cows with high BHB levels during the ketosis period were more likely to experience endometritis and mastitis [24,25]. Nevertheless, BHB has also been observed to possess anti-inflammatory qualities, which are in contradiction to our findings. For instance, it was demonstrated that BHB down-regulated the activation of the NF-κB pathway through GPR109A in BV2 cells [26]. Kambiz et al. also observed that nicotinic acid was able to diminish the cytokine production stimulated by GPR109A through the NF-κB signaling pathway in murine macrophages [27]. In rat pheochromocytoma cells, it was observed that the expression levels of p38, p-p38, and NF-κB decreased in a concentration-dependent manner when BHB was present in a 1 mM glucose environment [28]. It was also reported that BHB had a conflicting effect on the chemotaxis of monocytes. For example, Carretta et al. observed that bovine neutrophils exposed to GPR109A ligands (nicotinic acid and MK-1903), which imitate the presence of BHB, displayed enhanced chemotaxis [29], while another study indicated that dimethyl fumarate (a GPR109A agonist) inhibited the chemotaxis of neutrophils in mice [30]. The findings suggested that BHB could potentially have both pro-inflammatory and anti-inflammatory effects on cells from bovines. The varying role of BHB in oxidative stress in different species and tissues could be attributed to the differences observed [31]. Indeed, BHB has been reported to have a protective effect on HEK293 cells against oxidative stress caused by paraquat [32], yet it has been observed to induce oxidative damage in bovine endometrial cells [33]. Hence, it appears that the impact of BHB on the regulation of inflammation may be contingent on the type of cell and the environment. Our study revealed that the pro-inflammatory effects of low glucose plus BHB could be counteracted by inhibitors of the GPR109A/NF-κB signaling pathway (Figure 4 and Figure 5), indicating that low glucose plus BHB result in an upsurge of IL-1β, IL-6, and TNF-α expression levels in yak AMs through the activation of the GPR109A/NF-κB signaling pathway. As AMs are the most prevalent resident innate immune cells involved in the inflammatory process [7], it is likely that long-term starvation-induced low glucose and high BHB can elevate the risk of pneumonia through the activation of the GPR109A/NF-κB signaling pathway and the release of pro-inflammatory factors.

NF-κB is kept inactive in the cytoplasm until it is activated, at which point it is swiftly imported into the nucleus [34]. NF-κB p65 is acknowledged to be a crucial element in regulating inflammatory reactions due to its role in manufacturing various key pro-inflammatory factors [35]. BHB has been found to increase IKKβ activity, cause IκBα phosphorylation, and facilitate the movement of NF-κB p65 into the nucleus [17,19]. Activated p38 has been found to induce the expression of inflammatory cytokines, thus triggering the activation of NF-κB, a key mediator of the immune response [36]. It is believed that the NF-κB signaling pathway is the classic pro-inflammatory signaling pathway due to its ability to regulate IL-1β, IL-6, and TNF-α [37,38]. It has been previously reported that BHB was able to reduce the buildup of ROS in pheochromocytoma cells and further suppress apoptosis by modulating the p38 signaling pathway when glucose was lacking [28]. PPARγ is a transcription factor that has anti-inflammatory properties and can counter NF-κB-induced cytokine production in a continual manner [39]. Previous studies have confirmed that the expression level of *GPR109A* has been connected with PPARγ activation [40,41]. The results of this experiment suggest that BHB’s pro-inflammatory effects may be connected to the activation of p38 and the reduction in PPARγ expression, as primary AMs exposed to low glucose plus BHB saw a significant decrease in mRNA expression and phosphorylation levels of p38, while the total protein level of p38 stayed the same and mRNA and protein expression levels of PPARγ also decreased, leading to a decrease in the anti-inflammatory effect of PPARγ (Figure 6). Nevertheless, further investigation is necessary to understand the exact mechanism.

## 4. Materials and Methods

### 4.1. Yak Primary AMs Isolation and Culture

The isolation of the primary AMs was accomplished referring to a previous study [42]. Briefly, three intact lungs were removed from three freshly slaughtered, healthy Jiulong yaks (males, about 2.5 years old, with no pulmonary lesions), leaving 12 cm of trachea left on the lungs, which were washed using a phosphate buffer saline (PBS; Solarbio, Beijing, China) containing 200 IU/mL of penicillin and 200 μg/mL of streptomycin (PS; Gibco, Carlsbad, CA, USA). Following the injection of the PBS into the lungs, the lungs were massaged lightly and the lavage fluid was collected; this was performed three times. The collected broncho alveolar lavage fluid was filtered by passing a sterile gauze with a mesh size of 200 (70 µm) before being centrifuged at 1200 rpm for six minutes, and the cells in the lower layer were then collected. Following the collection of AMs, they were washed twice with the PBS before being resuspended in an adherent medium composed of 90% Dulbecco’s modified Eagle’s medium (DMEM; Solarbio, Beijing, China), 10% fetal bovine serum (FBS, Sangon, Shanghai, China), and 1% PS. After six hours of incubation, the non-adherent cells were washed with PBS and removed. Then, the left adherent cells were identified using a-naphthyl acetate esterase (Solarbio, Beijing, China) staining, and their viability was then determined using trypan blue exclusion. The cell density was changed to 1 × 10^6^, and subsequently the cells were inoculated into a six-well tissue culture plate (NEST, Wuxi, China) and incubated at 37 °C with 5% CO_2_. In accordance with a previous study [43], we stimulated AMs with 5.5 and 2.8 mM concentrations of glucose to replicate the normal- and low-glucose conditions experienced by yaks. In the experiments involving inhibitors, the inhibitors were dissolved in 100% dimethyl sulfoxide (DMSO; Sigma-Aldrich, Saint Louis, MO, USA) and then diluted in a culture medium, which was then used to pretreat the AMs (60–70% confluent) for 30 min. The concentrations of pertussis toxin (PTX; Tocris, Shanghai, China) and pyrrolidinedithiocarbamic (PDTC; MCE, Monmouth Junction, NJ, USA) to be used in the experiment were determined through a pre-experiment.

### 4.2. BHB Preparation

The BHB (Sigma-Aldrich, Dublin, Ireland) was dissolved in sterile deionized water to final concentrations of 1, 2, 4, 8, 16, and 32 mM and filtered with 0.22 μm membrane filtration sterilization. The solution was then stored at −20 °C for subsequent usage.

### 4.3. Cell Viability Assay

Following the complete attachment of the AMs, cells were treated with varying concentrations of BHB, and the viabilities of these cells were measured after 12 and 24 h of incubation using the cell counting kit-8 (CCK-8; Beyotime, Shanghai, China) according to its instruction. Briefly, approximately 10^5^ cells suspended in 0.1 mL of a medium were seeded to a 96-well plate. The plate was then incubated at 37 °C with 5% CO_2_ for 6 h. After cell adhesion, the medium was replaced with a new medium containing 0, 1, 2, 4, 8, 16, and 32 mM BHB, and then it was incubated for 12 or 24 h. Afterward, 10 μL of the CCK-8 solution was added to the well, and then the plate was incubated for three hours. At last, the absorbance value of each well at 450 nm was measured using a Micro Plate Spectrophotometer (TECAN, Männedorf, Switzerland).

### 4.4. Real-Time Quantitative Polymerase Chain Reaction (RT-qPCR)

Following the instructions, Trizol reagent (Gibco, Carlsbad, CA, USA) was used to extract total ribonucleic acid (RNA) from AMs in each treatment, and the quantity of the extracted RNA was estimated using a NanoDrop 2000 (Thermo, Waltham, MA, USA). The Prime Script^TM^ RT reagent Kit with gDNA Eraser (TaKaRa, Tokyo, Japan) was employed to generate complementary deoxyribonucleic acid (cDNA) following the manufacturer’s instructions. The expression levels of the considered mRNA were then quantified using the SYBR Green method (Takara, Tokyo, Japan) and a real-time thermocycler (Bio-Rad, Hercules, CA, USA), calculated using the 2^−∆∆Ct^ method and normalized relative to the expression levels of the housekeeping gene *β-actin*. A 10 µL system was used to carry out the amplification with a thermal cycle of 95 °C for 3 min, followed by 40 cycles of 95°C for 10 s and 58 °C for 30 s. The sequences of the primers employed were as follows: *β-actin*, Forward 5′-GCCCATCTATGAGGGGTACG-3′ and Reverse 5′-TCACGGACGATTTCCGCT-3′; *GPR109A*, Forward 5′-ACATCACCCTCAGCTTCACC-3′ and Reverse 5′-GCGGTTGTTATCCGACTCAT-3’; p38, Forward 5’-AAGTAGCCAGGTTGTCG-3′ and Reverse 5′-AGAGGAATGGCGATGA-3′; *p65*, Forward 5′-GAGATCATCGAGCAGCCCAA-3′ and Reverse 5′-ATAGTGGGGTGGGTCTTGGT-3′; and *PPARγ*, Forward 5′-CTGTGAAGTTCAACGCACTGGAATTAG-3′ and Reverse 5′-TGCAGCAGATTGTCTTGTATGTCCTC-3′.

### 4.5. Immunoblot Assay

Utilizing a RIPA lysis buffer (Solarbio, Beijing, China), the total cellular proteins of the AMs in each treatment were extracted and denatured for 10 min at 100 °C. Through the use of sodium dodecyl sulfate-polyacrylamide gels, an equivalent amount of protein was segregated and then shifted to a polyvinylidene fluoride membrane. Following incubation with primary antibodies, the membranes were hybridized with corresponding secondary antibodies (1:5000; #SE134; Solarbio, Beijing, China) and then further incubated with enhanced chemiluminescent reagents to make the protein bands visible. The primary antibodies used in this experiment included GPR109A (1:1000; #10076; Bioss, Beijing, China), p-p38 (1:1000; #4511; CST, Shanghai, China), p38 (1:1000; #8690T; CST, Shanghai, China), p65 (1:1000; #4764T; CST, Shanghai, China), PPARγ (1:800; #ab45036; Abcam, London, UK), and β-actin (1:5000; #ab8226; Abcam, London, UK). The band intensity of the images was measured using the Tanon-5200 automatic chemiluminescence imaging analysis system (Tanon, Shanghai, China).

### 4.6. Enzyme-Linked Immunosorbent Assay (ELISA)

The supernatant concentrations of IL-1β, IL-6, and TNF-α in each treatment were detected using corresponding commercialized ELISA kits (Nanjing Jiancheng, Nanjing, China) as per the manufacturer’s instructions.

### 4.7. Statistical Analyses

All experiments were replicated at least three times, and the data were expressed as the mean ± standard deviation. Statistical analysis was carried out using SPSS 26 (IBM, NY, USA), and graphs were prepared in GraphPad Prism 9.0 (GraphPad Software, La Jolla, CA, USA) and Adobe Illustrator 2022 (Adobe Systems Incorporated, San Jose, CA, USA). A Mann–Whitney U test or one-way ANOVA and Bartlett’s test were used to analyze the differences between and among groups. Differences were considered to be statistically significant when *p* < 0.05.

## 5. Conclusions

In summary, our study confirmed that the low glucose plus BHB condition would elevate the release of pro-inflammatory cytokines IL-1β, IL-6, and TNF-α in yak AMs by activating the GPR109A/NF-κB signaling pathway. Our findings may explain why yaks experience higher rates of respiratory diseases and mortality, thus offering new insight into the prevention and treatment of bovine respiratory diseases. Nevertheless, further research, particularly in vivo studies, is necessary to validate our findings.

## Figures and Tables

**Figure 1 ijms-24-11331-f001:**
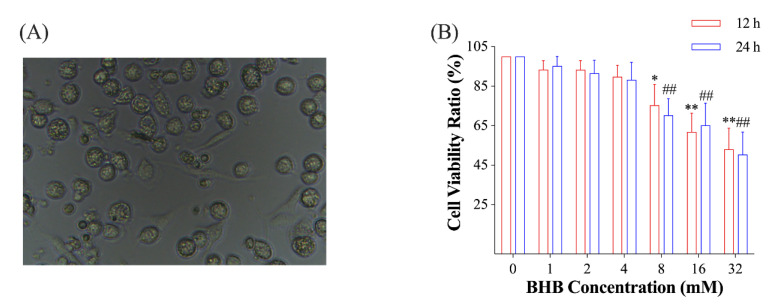
(**A**), a picture of the isolated yak primary AMs (400×); (**B**), the ratio of the viability of AMs treated with varying concentrations of BHB for 12 or 24 h to that in the control group (*n* = 3). In (**B**), data were expressed as the mean ± standard deviation. One-way ANOVA was used to analyze the differences among groups. * 0.01 < *p* < 0.05 for AMs treated for 12 h; ** 0.001 < *p* < 0.01 for AMs treated for 12 h; ## 0.001 < *p* < 0.01 for AMs treated for 24 h.

**Figure 2 ijms-24-11331-f002:**
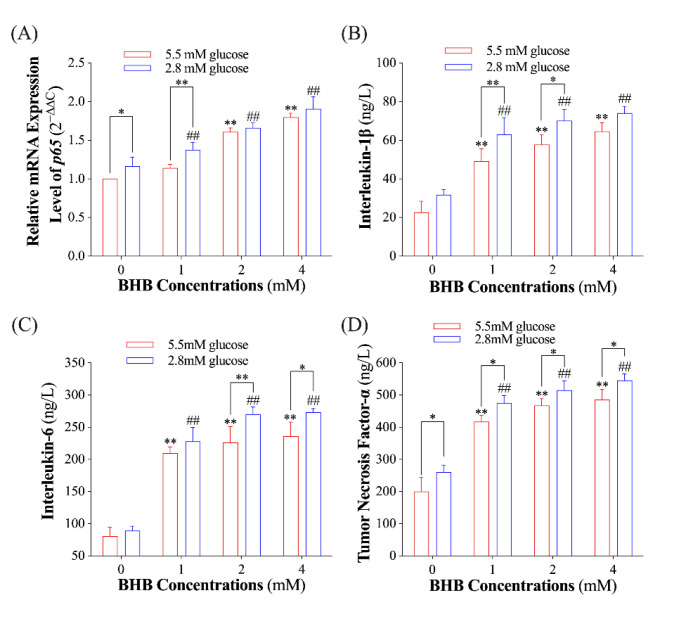
The effect of BHB on the pro-inflammatory response of AMs in normal- and low-glucose conditions. (**A**), the relative mRNA expression level of the *p65* gene normalized to *β-actin* (*n* = 6); (**B**–**D**), the supernatant concentrations of interleukin-1β, interleukin-6, and tumor necrosis factor-α in AMs exposed to normal/low glucose plus varying concentrations of BHB (*n* = 5). Data were expressed as the mean ± standard deviation. One-way ANOVA was used to analyze the differences among groups. * 0.01 < *p* < 0.05 for AMs treated with normal glucose; ** 0.001 < *p* < 0.01 for AMs treated with normal glucose; ## 0.001 < *p* < 0.01 for AMs treated with low glucose.

**Figure 3 ijms-24-11331-f003:**
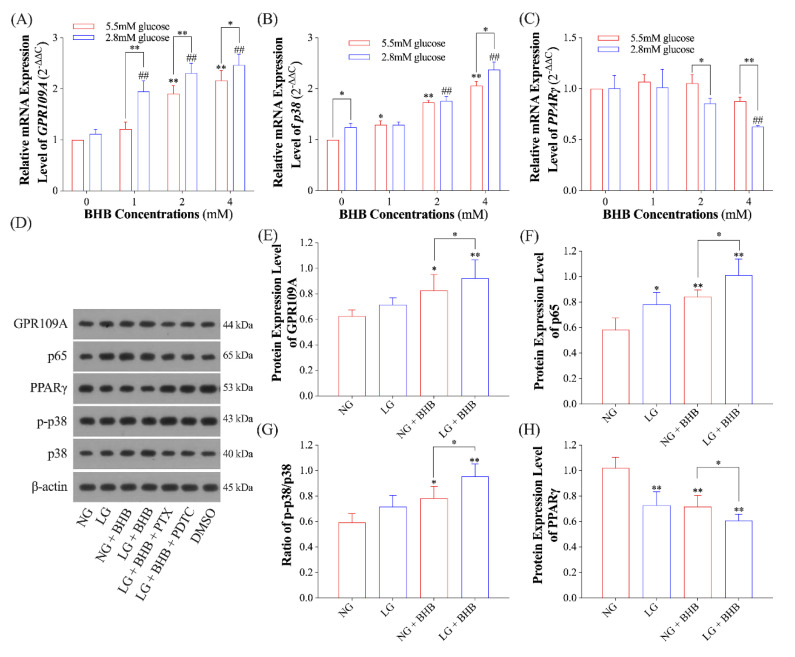
Effects of low glucose plus BHB on the GPR109A/p38/NF-κB signaling pathway. (**A**–**C**), the relative mRNA expression levels of *GPR109A*, *p38*, and *PPARγ* (*n* = 6); (**D**–**H**), the gel picture and grayscale analysis of the immunoblot assay of GPR109A, p65, PPARγ, p-p38, and p38 (*n* = 3). Data were expressed as the mean ± standard deviation. One-way ANOVA was used to analyze the differences among groups. In (**A**–**C**): * 0.01 < *p* < 0.05 for AMs treated with normal glucose; ** 0.001 < *p* < 0.01 for AMs treated with normal glucose; ## 0.001 < *p* < 0.01 for AMs treated with low glucose. In (**E**–**H**): * 0.01 < *p* < 0.05; ** 0.001 < *p* < 0.01. NG, normal glucose; LG, low glucose.

**Figure 4 ijms-24-11331-f004:**
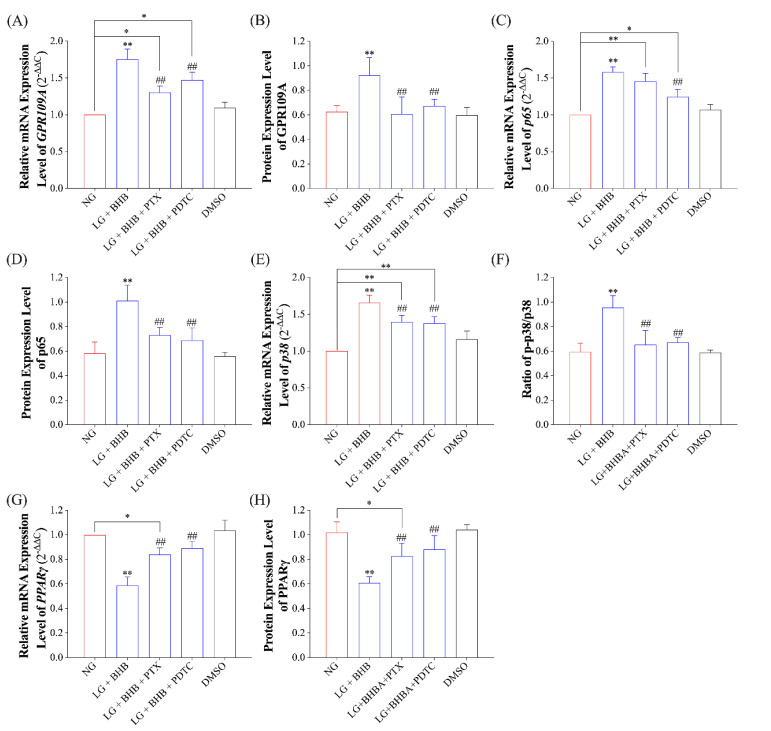
PTX and PDTC treatment blocked the GPR109A/p38/NF-κB signaling pathway. (**A**,**C**,**E**,**G**), the relative mRNA expression levels of *GPR109A*, *p65*, *p38*, and *PPARγ* (*n* = 6); (**B**,**D**,**H**), the grayscale analysis of the immunoblot assay of GPR109A, p65, and PPARγ (*n* = 3); (**F**), the ratio of p-p38 to p38 in AMs from each group (*n* = 3). Data were expressed as the mean ± standard deviation. One-way ANOVA was used to analyze the differences among groups. * 0.01 < *p* < 0.05 vs. NG; ** 0.001 < *p* < 0.01 vs. NG; ## 0.001 < *p* < 0.01 vs. LG + BHB. NG, normal glucose; LG, low glucose.

**Figure 5 ijms-24-11331-f005:**
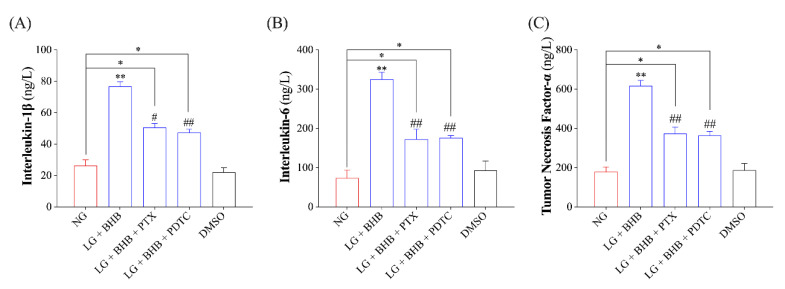
Inhibition of the GPR109A/p38/NF-κB signaling pathway blocked the release of pro-inflammatory cytokines induced by low glucose plus BHB. (**A**–**C**), supernatant concentrations of interleukin-1β, interleukin-6, and tumor necrosis factor-α in AMs from each treatment (*n* = 5). Data were expressed as the mean ± standard deviation. One-way ANOVA was used to analyze the differences among groups. * 0.01 < *p* < 0.05 vs. NG; ** 0.001 < *p* < 0.01 vs. NG; # 0.01 < *p* < 0.05 vs. LG + BHB; ## 0.001 < *p* < 0.01 vs. LG + BHB. NG, normal glucose; LG, low glucose.

**Figure 6 ijms-24-11331-f006:**
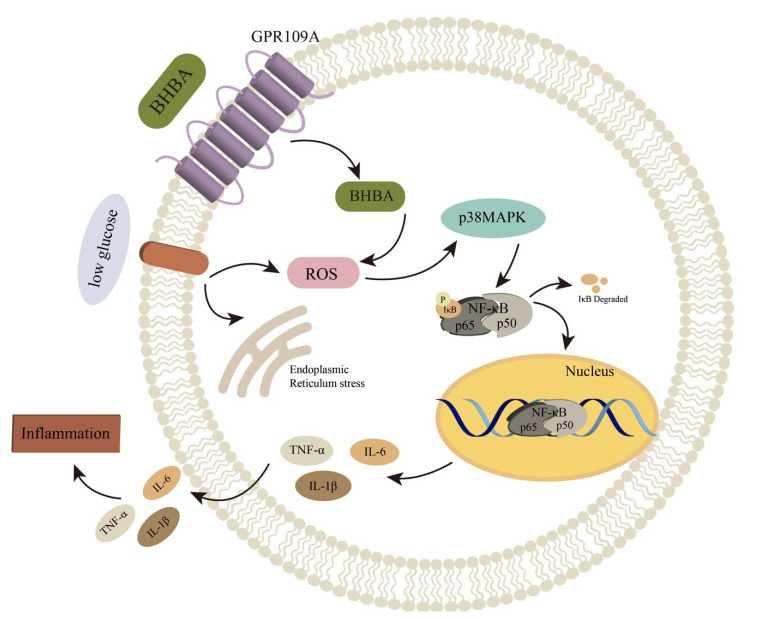
The possible mechanisms of the pro-inflammatory response in yak AMs induced by low glucose plus BHB. In the condition of low glucose plus BHB, the GPR109A/NF-κB signaling pathway is activated through the receptor GPR109A. Oxidative stress is induced, which activates p38, which consequently stimulates NF-κB and results in the expression and release of inflammatory cytokines. In addition, low glucose can also induce oxidative stress and endoplasmic reticulum stress in the AMs.

## Data Availability

The data used to support the findings of this study are available from the corresponding authors upon request.

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
