# Peer review of "Low Glucose plus β-Hydroxybutyrate Induces an Enhanced Inflammatory Response in Yak Alveolar Macrophages via Activating the GPR109A/NF-κB Signaling Pathway"

_ijms, 2023, doi:10.3390/ijms241411331_

Round 1

Reviewer 1 Report

Dear Colleagues,

Thank you for interesting and original work.  Only one minor comment -- please indicate number of replicates on each plot. 

Best regards,

Reviewer 2 Report

This paper investigated basic immunobiology of primary yak alveolar macrophages.

The authors clearly showed that the macrophages cultured in a low glucose condition (hypoglycemia) produced pro-inflammatory mediators (IL-6 TNF ). Furthermore, they also showed that the cells stimulated by BHB, a short-chain fatty acid, induced such inflammatory mediators dose dependent manner and synergistically enhanced the inflammatory responses induced low-glucose condition.

From pharmacological approaches using a NF-κB inhibitor and PTX (a G-protein inhibitor), they confirmed that those inflammatory responses were induced by NF-κB signaling pathway and signaling pathways of G-protein coupled receptor family (GPR109A, a main receptor for butyrate, is one candidate).

The authors concluded that GPR109A/NF-κB signaling pathway might contribute to induce pro-inflammatory responses in primary yak alveolar macrophages cultured with low-glucose and BHB.

The concept of this study and data are interesting, however, some supportive data or information are required to establish.

1)At the result part of the manuscript, the line 64, authors mentioned that “As the main receptor of BHB, GPR109A participates in the regulation of the phenotype and function of macrophages [14]”, however, the ref. paper did not use BHB or Butyrate and also did not show the direct evidence of BHB as the main ligand for GPR109A. Authors should replace the ref. paper or modify to the adequate sentence.

2) To improve the novelty of this manuscripts and authors study, we recommend adding some physiological information of yacs. We recommend adding information of the serum level of glucose and BHB of normal and starved yacs. Moreover, it should be possible for authors to explain that the situation of yac alveolar macrophages subjected to 2.8 mM to 5.5 mM of glucose and 0-4 mM of BHB is reasonable.

3) In the figure 4 and 5, authors used PTX as GPR109A inhibitor, however the inhibitor, which has a broad spectrum in inhibition of G-protein coupled receptors, may affect not only GPR109A but also other receptors. We recommend adding direct evidence of GPR109A as a main receptor of BHB in the macrophages. At least, authors should site the precise paper at the discussion part of the manuscript.

4) In the figure 6, TLR2 and 4 seem to associate with BHB. Authors should discuss at appropriate area of this manuscript.

Reviewer 3 Report

In the paper under review the authors devoted to finding out the reasons for the high prevalence of respiratory diseases in yaks during the off-season. The work was performed in the in vitro system on AM of bronchoalveolar lavage of yaks. The results are clearly stated. However, there are a few minor remarks.

line 90-92 In order to assess the ideal concentration and duration of BHB 90 exposure to the AMs, we administered 1, 2, 4, 8, 16, and 32 mM of BHB to the AMs for a 91 period of either 12 or 24 hours under normal glucose condition. Authors should indicate the concentration of glucose, because previously there was no reference to the norm.

line 99 picture of the isolated yak primary AMs (40 ×) As far as I understood authors mean the lens magnification, while eyepiece magnification varies from microscope to microscope. Articles usually give true magnification (lens magnification * eyepiece magnification). Please provide true magnification.

line 286-290 The authors should explain principle of the CCK-8 method.

Round 2

Reviewer 2 Report

The authors responded to my comments logically and cordiality. Thank you.